# Giant Hepatic Artery Aneurysm

**DOI:** 10.3390/diagnostics9020053

**Published:** 2019-05-13

**Authors:** Farid Gossili, Helle D. Zacho

**Affiliations:** 1Department of Nuclear Medicine, Aalborg University Hospital, Hobrovej 18-22, DK-9000 Aalborg, Denmark; h.zacho@rn.dk; 2Department of Clinical Medicine, Aalborg University, DK-9000 Aalborg, Denmark

**Keywords:** ^18^F-Fludeoxyglucose positron emission tomography/computed tomography (^18^F-FDG PET/CT), giant hepatic artery aneurysm (HAA)

## Abstract

Hepatic artery aneurysm (HAA) is the second most common type of visceral aneurysm. Giant HAAs (larger than 5 cm) are very rare. We present a case of an asymptomatic giant hepatic artery aneurysm (diameter 10.7 cm) discovered as an incidental finding on an ^18^F-fludeoxyglucose positron emission tomography/computed tomography (^18^F-FDG PET/CT) scan of a patient admitted for pretreatment staging of urothelial carcinoma.

## 1. Introduction

Visceral artery aneurysms comprise 1–2% of all vascular diseases [1]. Based on post-mortem reports, their prevalence has been estimated to be approximately 10% [2], which is higher than that of abdominal aortic aneurysms (AAAs). However, the actual prevalence of splanchnic aneurysms is unknown [3]. Hepatic artery aneurysm (HAA) is the second most common type of visceral aneurysm, following splenic artery aneurysm. The prevalence of HAAs varies from 0.002% [3,4] to 0.4% [4,5] of all arterial aneurysms and from 12% [3] to 20% of splanchnic aneurysms [1,5,6,7,8,9].

In the majority of studies, the prevalence of HAAs accounts for both true and false aneurysms, but only a few studies distinguish between true aneurysms and pseudo-aneurysms. The majority of HAAs are true aneurysms that are extrahepatic and single [3,10,11]. True extrahepatic aneurysms are often comorbid with hypertension, followed by malignancy and peripheral vascular disease [3,11]. Most HAAs are asymptomatic and are thus diagnosed incidentally in the course of imaging for other diseases [10]. The prevalence of incidental finding of HAAs by computed tomography (CT) based on population studies has not been reported, whereas the prevalence of incidental finding of AAAs in radiological studies including CT and magnetic resonance imaging (MRI) is reported to be between 0.5 and 2.2% [12,13]. Hybrid imaging studies, including positron emission tomography/computed tomography (PET/CT), are often used in the staging and restaging of cancer patients, and the prevalence of incidental aneurysms (all types) in PET/CT has been reported to be 1.7% [14] However, early diagnosis and initiation of therapy is important because there is a high risk of rupture and intraabdominal bleeding [10].

Giant HAAs (larger than 5 cm) are extremely rare, and very few cases have been reported [8,15,16,17]. Moreover, the true incidence of giant visceral aneurysms is unknown. Although no relationship between aneurysm size and risk for rupture has been shown in visceral aneurysms [11,17], several studies recommend treatment for aneurysms larger than 5 cm to eliminate the risk of rupture [3].

We present a case of a true, extrahepatic giant HAA, which was detected by ^18^F-fludeoxyglucose positron emission tomography/computed tomography (^18^F-FDG PET/CT) and confirmed with Doppler ultrasound.

## 2. Case Report

A previously healthy, 72 year old man with a history of well-regulated hypertension was diagnosed with bladder cancer causing severe nephropathy due to bilateral obstruction of the ureter ostium. The patient was referred to an ^18^F-FDG PET/CT for staging. ^18^F-FDG PET/CT was acquired 60 min after injection of 280 MBq ^18^F-FDG on a Biograph mCT Flow 64 (Siemens Medical Solutions, Erlangen, Germany) with a flow table of 0.8 mm/s. The PET images were reconstructed using Ordered Subset Expectation Maximum, applying time-of-flight and point-spread-function. A CT (120 kV/10–150 mA) was performed without intravenous contrast due to the patient’s nephropathy.

Non-enhanced CT showed a large hollow process with a distinct calcified wall mimicking a porcelain gallbladder. However, the fused PET/CT demonstrated a discrete FDG uptake in the central part of the process, comparable to blood pool activity (SUVmax 3.5) with a surrounding mural thrombus (Figure 1) corresponding to a large HAA measuring approximately 107 × 105 mm with a thick calcified wall. Color Doppler ultrasound was performed to confirm the diagnosis. Ultrasound showed a giant aneurysm with perimural thrombus and intraluminal turbulent flow (Figure 2). In addition, ^18^F-FDG PET/CT revealed lymph node metastases and peritoneal carcinosis.

The patient was discussed at the urological multi-disciplinary team conference and, based on the disseminated urothelial carcinoma, no treatment with curative intent was possible. The patient was offered palliative chemotherapy. For evaluation and potential treatment of the giant HAA, the patient was referred to the Department of Vascular Surgery. Based on the poor prognosis of the cancer, it was decided that there was no indication for either open or endovascular treatment of the aneurysm due to the high risk of such procedures. Palliative treatment was initiated for the bladder cancer, and the patient was followed with non-contrast enhanced CT.

A total of three CT scans were performed, showing minimal progression of the HAA from 107 × 105 mm to 110 × 106 mm. The patient did not experience any symptoms related to the aneurysm until his death 8 months later. The patient presented in this case report was identified during a quality assurance of our use of FDG PET/CT in bladder cancer. This study was approved by the Danish National Data Protection Agency and according to national legislation retrospective, observational studies do not require approval from the ethical committee.

## 3. Discussion

Owing to the increasing availability of imaging modalities over the last decades, the number of aneurysms detected prior to rupture has increased [5,7]. This increase provides the opportunity to fully evaluate patients and consequently choose the most effective form of management for each individual patient. The early diagnosis of hepatic artery aneurysm is crucial because the majority of HAAs are asymptomatic before rupture [5,17]. With the widespread use of hybrid studies such as PET/CT, incidental findings of aneurysms have been reported. PET/CT is usually used in the diagnosis, staging, and restaging of cancer patients, but incidental findings by PET/CT may have a high clinical significance, owing to the combination of functional and anatomical data [14]. Incidental findings of HAAs by PET/CT are, however, rarely reported, but the present case is not an exception [5]. In this study, the giant HAA was initially simulating porcelain gallbladder in a non-contrast enhanced CT, but the FDG uptake in the central part of the lesion, comparable with blood pool activity, raised the concern of the giant HAA in PET/CT.

Although the gold standard for diagnosing HAA is angiography [5,15,18,19], other diagnostic modalities such as CT-angiography, magnetic resonance angiography (MRA), and particularly Doppler ultrasound have become the first choice for the final diagnosis of HAA owing to the non-invasive nature of these techniques [7].

Doppler ultrasound is convenient for immediate confirmation of vascular lesions, determination of exact location and size of aneurysms, and differentiation from other vascular lesions, such as arteriovenous fistulas or vascular malformations [15]. Doppler ultrasound is often the first choice for the assessment of aneurysmal disease, particularly in patients with renal failure [15]. This was also the case in our patient, where the final diagnosis was performed by Doppler sonography.

The exact risk of rupture of a visceral aneurysm is unknown but has been reported to be between 14% and 80% [1,11]. The presence of multiple aneurysms and non-atherosclerotic etiology have been described as risk factors for the rupture of HAA [2]. Nonetheless, no relationship between the size of HAA and rupture has been described [2,3,11].

According to the Society of Vascular Surgery (www.vascular.org), no formal guidelines regarding visceral artery aneurysms exist, though some are under development. In the available literature, Abbas et al. [3] suggested that aneurysms larger than 2 cm should be treated in patients fit for surgery who had a life expectancy of more than 2 years. However, patients with multiple comorbidities and an aneurysm in the range of 2–5 cm can undergo regular follow-up without intervention; finally, HAAs larger than 5 cm should generally be treated with open surgery, embolization, or ligation. Pasha et al. [2] recommend intervention in symptomatic patients regardless of aneurysm size, due to the increased risk of rupture. Such recommendations seem sound compared to treatment recommendations for abdominal aortic aneurisms. However, to the best of our knowledge, no studies on the conservative treatment of giant HAAs have been published. For HAAs smaller than 5 cm, Pasha et al. conducted follow-up without intervention in 22 patients with HAA (mean diameter 2.3 cm; range 1.5–5 cm) for an average of 68.4 months (range 1–372 months). No ruptures were identified [2].

In the present case, considering the patient’s short life expectancy, a conservative treatment course was adopted, and the patient’s aneurysm remained almost unchanged for 8 months.

This present case highlights the ability of PET/CT to identify visceral aneurysms and to influence the diagnostic strategy and further treatment. Furthermore, due to the very rare frequency of giant HAAs and the absence of guidelines endorsed by internal societies of vascular surgery, the necessity of collecting multi-center data for the purpose of optimizing the management of patients with giant HAA should be addressed in the future.

## Figures and Tables

**Figure 1 diagnostics-09-00053-f001:**
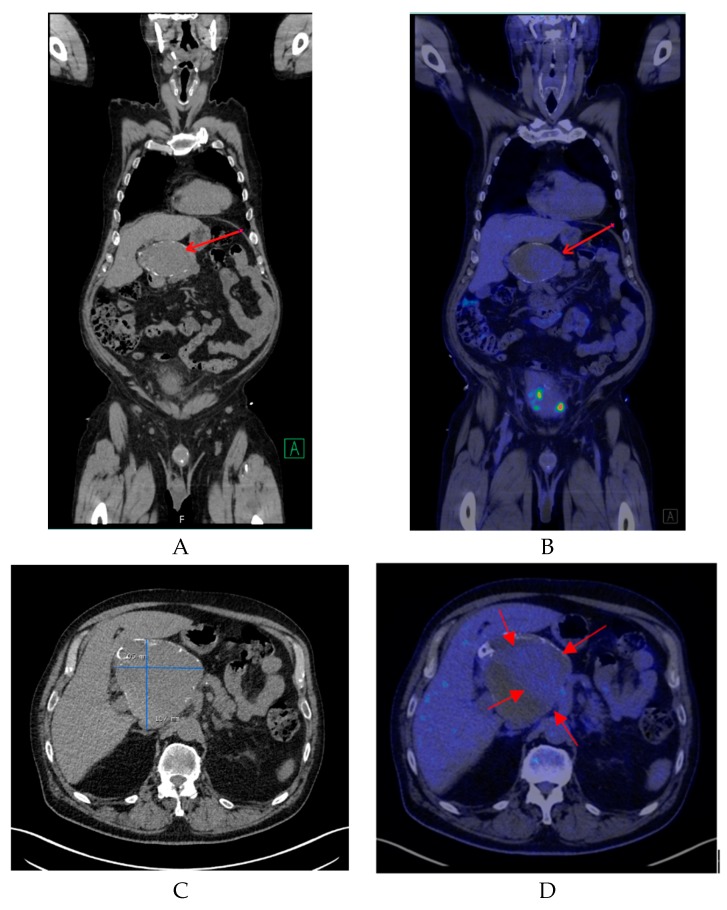
(**A**) A coronal non-contrast enhanced computed tomography (CT) image of the lesion (red arrow). (**B**) A coronal fused positron emission tomography/computed tomography (PET/CT) image showing the same region with mild ^18^F-FDG uptake (red arrow). (**C**) A transverse non-contrast CT image confirming the presence of a cyst-like lesion with a hyperdense wall at the hepatic hilum, measuring approximately 107 × 105 mm. (**D**) A transverse fused PET/CT image demonstrating mild hypermetabolic activity in the center of the lesion compatible with blood pool activity (red arrows) and surrounding mural thrombus.

**Figure 2 diagnostics-09-00053-f002:**
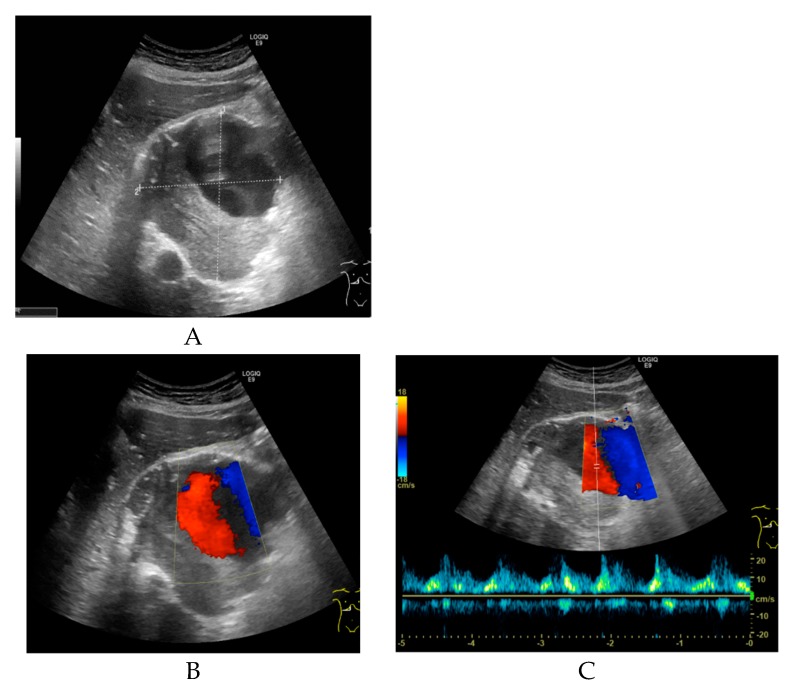
(**A**) Transverse grey-scale ultrasound of the porta hepatis reveals a large, well-defined sonolucent mass measuring approximately 9.7 × 9.4 × 9.1 cm. The upper part of the lesion shows an anechoic lumen, and the lower part of the lesion contains intraluminal echogenic material suggestive of thrombus. (**B**) Transverse color Doppler ultrasound shows a turbulent arterial flow pattern in the upper part of the lesion. (**C**) Pulsed wave Doppler sonogram shows an arterial waveform with high peak systolic velocity consistent with a hepatic artery aneurysm.

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
