# Peer review of "Giant Hepatic Artery Aneurysm"

_diagnostics, 2019, doi:10.3390/diagnostics9020053_

Round 1
Reviewer 1 Report
It is a simple case report. The case is nicely described by the authors. However I am not abundantly sure about the real value of this case report for the scientific community. The authors should clarify the real message of their paper. Moreover, in the conclusion the authors state that "This present case highlights the necessity of conducting multi-centre studies for the purpose of optimizing the management of patients with aneurysms". What does it mean? Why does the simple description of a large aneurysm accidentally found at PET/CT imaging should support the design of multicentre studies? To what aim?Author Response
1.It is a simple case report. The case is nicely described by the authors. However, I am not abundantly sure about the real value of this case report for the scientific community. The authors should clarify the real message of their paper.
Answer: Thanks for the comment you provided. We agree that the main messages of this study is not fully explained, therefore, we have clarified this in the introduction p.1 line 13-20 and elaborated on this topic in the Discussion (p.4, line 6-13).
2. Moreover, in the conclusion the authors state that "This present case highlights the necessity of conducting multi-centre studies for the purpose of optimizing the management of patients with aneurysms". What does it mean? Why does the simple description of a large aneurysm accidentally found at PET/CT imaging should support the design of multicentre studies? To what aim?
Answer: The authors fully agree with the reviewer, that this sentence is not entirely appropriate. What was meant was; that due to the low incidence of HAA, data on treatment vs non-treatment has to be gathered from multiple centers in order to evaluate the recommendations and to develop guidelines in this field. This sentence has been modified in agreement with the reviewer (p.5, line 14-17).
Reviewer 2 Report
In the paper entitled “Giant Hepatic Artery Aneurysm” the authors describe a case report with an incidental finding of a Giant HAA on an 18F-FDG PET/CT. In my personal opinion, considering the rare occurrence of such disease/condition this manuscript represents a case report that should be published and disseminated across the scientific community. In addition, it provides the sufficient background for the discussion of the management of patients with aneurysms when with other complications (such as terminal oncological disease). However, in such context, I would like to suggest that minor revisions could be undertaken to improve the scientific value of the manuscript.
(1. Introduction) The second paragraph could be reviewed in order to highlight (if possible, with quantitative information) the typical incidental diagnosis of such aneurysms.
(2. Case report) 18F-FDG PET/CT technical protocol should be mentioned (injected activity of the radiopharmaceutical, image acquisition parameters, …) in order to validate the usefulness of the image modality in such context – not to be considered as a diagnostic method, but to disseminate typical appearance of the disease in images obtained with such parameters. If possible, SUV values for the lesion should be given to disseminate typical 18F-FDG uptake patterns.
(3. Discussion) PET/CT examination is never cited during the discussion. As it was the image modality responsible for the occasional finding, it should be mentioned and reviewed in order to evaluate the existence of published data about similar findings in the same modality. Even if not totally related with this case report, some published papers refer to the use of PET/CT (with 18F-FDG or other radiopharmaceuticals) for the evaluation and/or follow-up of patients with abdominal aneurysms.
Author Response
In the paper entitled “Giant Hepatic Artery Aneurysm” the authors describe a case report with an incidental finding of a Giant HAA on an 18F-FDG PET/CT. In my personal opinion, considering the rare occurrence of such disease/condition this manuscript represents a case report that should be published and disseminated across the scientific community. In addition, it provides the sufficient background for the discussion of the management of patients with aneurysms when with other complications (such as terminal oncological disease). However, in such context, I would like to suggest that minor revisions could be undertaken to improve the scientific value of the manuscript.
1. Introduction: The second paragraph could be reviewed in order to highlight (if possible, with quantitative information) the typical incidental diagnosis of such aneurysms.
Answer: Thank you for your comment. The change has conducted in accordance with the reviewers’ suggestion (Abstract, p.1, line 3, and in the text p1, line 13-18 and added reference number 12-14)
2. Case report: 18F-FDG PET/CT technical protocol should be mentioned (injected activity of the radiopharmaceutical, image acquisition parameters, …) in order to validate the usefulness of the image modality in such context – not to be considered as a diagnostic method, but to disseminate typical appearance of the disease in images obtained with such parameters. If possible, SUV values for the lesion should be given to disseminate typical 18F-FDG uptake patterns.
Answer: The details regarding the 18F-FDG PET/CT protocol have been added to the manuscript in agreement with the reviewer (p2, line 11-14).
3. If possible, SUV values for the lesion should be given to disseminate typical 18F-FDG uptake patterns.
Answer: Solved. SUV has been added (p.2, line 17).
4. Discussion: PET/CT examination is never cited during the discussion. As it was the image modality responsible for the occasional finding, it should be mentioned and reviewed in order to evaluate the existence of published data about similar findings in the same modality. Even if not totally related with this case report, some published papers refer to the use of PET/CT (with 18F-FDG or other radiopharmaceuticals) for the evaluation and/or follow-up of patients with abdominal aneurysms.
Answer: We have added a paragraph in p.2, line 15-18 and in the Discussion (p.4, line 11-15) regarding the contribution of PET/CT in the diagnosis of the HAA agreement with the reviewer’s suggestion.
Reviewer 3 Report
Comments on the paper of Gossiii and Zacho.
Nice case.
Add some more info of the FDG PET balie, that is unclear now in section Introduction and section Discussion. What did PET add?
Also quantify FDG uptake by a SUVmax number.
Author Response
1.Nice case. Add some more info of the FDG PET balie, that is unclear now in section Introduction and section Discussion. What did PET add?
Answer: Thank you for pointing this out, we have elaborated on the contribution of PET/CT on p 2, line 15-18 and in the discussion section p.4, line 11-15 in agreement with the reviewers suggestion.
2. Also quantify FDG uptake by a SUVmax number
Answer: Solved. SUVmax has been added (p.2, line 17).
Round 2
Reviewer 1 Report
The article can be accepted in the present form.